# Modification of the Logistic Map Using Fuzzy Numbers with Application to Pseudorandom Number Generation and Image Encryption

**DOI:** 10.3390/e22040474

**Published:** 2020-04-20

**Authors:** Lazaros Moysis, Christos Volos, Sajad Jafari, Jesus M. Munoz-Pacheco, Jacques Kengne, Karthikeyan Rajagopal, Ioannis Stouboulos

**Affiliations:** 1Laboratory of Nonlinear Systems—Circuits & Complexity (LaNSCom), Physics Department, Aristotle University of Thessaloniki, 54124 Thessaloniki, Greece; lmousis@physics.auth.gr (L.M.); stouboulos@physics.auth.gr (I.S.); 2Nonlinear Systems and Applications, Faculty of Electrical and Electronics Engineering, Ton Duc Thang University, Ho Chi Minh City 758307, Vietnam; sajad.jafari@tdtu.edu.vn; 3Faculty of Electronics Sciences, Autonomous University of Puebla, Puebla 72000, Mexico; jesusm.pacheco@correo.buap.mx; 4Department of Electrical Engineering, University of Dschang, Dschang P.O. Box 134, Cameroon; kengnemozart@yahoo.fr; 5Center for Nonlinear dynamics, Defence University, Mekelle 1020, Ethiopia; rkarthiekeyan@gmail.com; 6Institute of Energy, Mekelle University, Mekelle 6330, Ethiopia

**Keywords:** chaos, logistic map, bifurcation analysis, fuzzy numbers, RBG, image encryption

## Abstract

A modification of the classic logistic map is proposed, using fuzzy triangular numbers. The resulting map is analysed through its Lyapunov exponent (LE) and bifurcation diagrams. It shows higher complexity compared to the classic logistic map and showcases phenomena, like antimonotonicity and crisis. The map is then applied to the problem of pseudo random bit generation, using a simple rule to generate the bit sequence. The resulting random bit generator (RBG) successfully passes the National Institute of Standards and Technology (NIST) statistical tests, and it is then successfully applied to the problem of image encryption.

## 1. Introduction

The field of chaos theory expands to numerous applications related to cryptography, secure communications, engineering, physics, economics, robotics, control, and many more; see, for example, References [1,2,3,4,5] and the references therein. Chaotic systems, being deterministic systems with a high sensitivity to initial conditions and parameter changes, constitute an excellent basis for efficiently modelling complex natural phenomena, as well as adding complexity to security related applications.

Due to the aforementioned applicability of chaotic systems, there is an ongoing demand for introducing novel chaotic systems. This is usually done by considering an existing chaotic system and modifying it, either by slightly altering a term in the system’s differential/difference equations, or by adding more nonlinear terms, or even by adding more variables and changing the system to a higher dimension.

The logistic map [6] is one of the most well-known one-dimensional discrete time chaotic systems and one of the most heavily modified chaotic systems; see, for example, References [7,8,9,10,11,12,13,14,15,16,17]. The map has only one parameter and a simple structure, which makes it suitable for many applications. In this work, we propose a modified version of the classic logistic map by employing fuzzy triangular numbers to modify its behavior. The idea of passing the values of the logistic map through a fuzzy number is mathematically very simple, yet it leads to a significant improvement of the chaotic behavior of the map, with many chaos related phenomena appearing, like antimonotonicity and crisis. Fuzzy logic and fuzzy sets are themselves a large field of study with innumerable applications in engineering and more [18,19,20]. Specifically in dynamical systems, fuzzy sets have been combined with chaotic systems in fuzzy dynamical systems; see, for example, References [15,21,22,23,24,25,26,27,28]. In the proposed modification of the logistic map, the values of the map in each iteration are passed through a triangular fuzzy number, which is a simple linear function that takes values on the interval [0,1]. The resulting map presents more complex chaos related phenomena compared to the classic map as mentioned above, as well as achieves a higher value for its Lyapunov exponent. Furthermore, to showcase the applicability of the map to chaos related applications, the problems of pseudo random bit generation [4,5,10,11,13,29,30,31,32,33,34,35,36,37,38,39,40,41,42,43] and image encryption [4,7,9,14,15,33,35,39,44,45,46,47,48] are considered. It is seen that the bit sequence generated from the modified map using a simple rule passes all 15 tests of the National Institute of Standards and Technology (NIST) statistical test suite [49]. The bit sequence generated is then applied to the problem of image encryption, and the resulting encrypted image is analysed for security using methods like histogram analysis, correlation, and information entropy.

It is important to note that this approach can be easily applied to any other one-dimensional chaotic system, as well as further modified by considering different types of fuzzy numbers, like trapezoidal, Gaussian, quadratic, exponential, or their combination. Thus, it is our belief that this approach for modifying a chaotic system will lead to more interesting works in the future.

The rest of the work is structured as follows: Section 2 presents some preliminaries on fuzzy numbers and the logistic map. In Section 3, the modified logistic map is proposed and its dynamical behavior is explored. Section 4 studies the application of the map in random number generation. The use of the produced bit sequence to the problem of image encryption is presented in Section 5. Finally, Section 6 concludes the paper with a discussion on future works.

## 2. Mathematical Preliminaries

### 2.1. Fuzzy Numbers

This section presents some preliminaries on fuzzy numbers; see [18,19,20] for a full presentation. There are slightly varying definitions for a fuzzy number, but in this work we consider a fuzzy number as a function f:X→[0,1] defined over a set X⊆R, such that:It is a normal fuzzy set, that is, there exists at least one x∈X, such that f(x)=1.Its a-cuts $f^{a}=\left\{x,\;f\left(x\right)\geq a\right\}$ are closed intervals ∀a∈[0,1].*f* is piecewise continuous.

Note that some works define fuzzy numbers as having exactly one $x_{0}$, such that $f(x_{0})=1$.

The fuzzy numbers considered here are defined as f:[0,1]→[0,1] and have the following triangular form:(1)$$f_{z}\left(x\right)=\left\{\begin{array}{cc} \frac{x}{z}, & \;0\leq x\leq z\\ \frac{1-x}{1-z}, & \;z\leq x\leq 1, \end{array}\right.$$
where *z* denotes the peak of the triangular fuzzy number. Examples for different values of *z* are shown in Figure 1. For example, for the fuzzy number $f_{0.5}\left(x\right)$, it holds that $f_{0.5}\left(0.4\right)=0.8$, which can be interpreted as the number 0.4 being equal to 0.5 with truth index 0.8.

### 2.2. The Logistic Map

The logistic map [6] is one of the most well-known discrete time 1D chaotic systems with a single parameter, described by
(2)$$x_{i+1}=rx_{i}\left(1-x_{i}\right),\;i=0,1,\ldots ,$$
where *r* is the system parameter that takes values in the interval [0,1], as well as $x_{0}\in \left[0,1\right]$. For different values of the parameter *r*, the following three dynamics are observed:For r<1, *x* decays to a fixed point x→0.For 1≤r≤3, the previous point loses its stability and another fixed point appears x=1/r.For 3≤r≤4 the system exhibits a rich behavior, going to chaos following a period doubling route.

Figure 2 shows the bifurcation diagram of the logistic map, and Figure 3 shows the diagram of its Lyapunov exponent, given by [50].
(3)$$LE=\lim_{n\to \infty }\frac{1}{n}\sum_{i=1}^{n}\ln\left|f^{\prime }\left(x_{i}\right)\right|,$$
which confirms the aforementioned dynamical behavior for various values of parameter *r*.

## 3. Implementation of Fuzzy Numbers to Logistic Map

We propose the following fuzzy number logistic map, described by
(4)$$x_{i+1}=f_{z}\left(rx_{i}\left(1-x_{i}\right)\right),$$
where *r* the bifurcation parameter, and $f_{z}$ denotes the fuzzy triangular function described above, centered at *z*. Thus, this modified logistic map has two parameters that can be tuned to affect the system’s behavior.

To unmask the dynamical behavior of the system, its bifurcation diagram is plotted with respect to parameter *r*. Examples for various values of *z* are given in Figure 4, Figure 5, Figure 6, Figure 7 and Figure 8, where many interesting phenomena are observed. First, it is easily seen in Figure 4 that for the two end values z=0 and z=1, the behavior of the system is very similar to the classic logistic map, having periodic behavior for values up until around 3.7, and then traversing to chaos through a period doubling route. For intermediate values of the parameter *z*, the system exhibits chaotic behavior for a range of parameter values *r*. The phenomenon of period doubling route to chaos appears in all cases, but also crisis phenomena are observed, where the system exits from chaos abruptly. The chaotic behavior is verified by the corresponding diagram of the Lyapunov Exponent (LE) of the system. Note that for z=0 and z=1, the diagram of the LE is the same as with the classic logistic map.

It is also very important to note that in many cases the Lyapunov exponent achieves a maximum value that is higher than 1, which is a lot higher than the value of the classic logistic map, which achieves a maximum value of around 0.7. This is clearly seen in Figure 9, which depicts the Lyapunov exponents of the system for various parameter values.

Similarly, considering the bifurcation diagrams of the system (Equation 4) with respect to parameter *z* unveils even more interesting phenomena, as seen in Figure 10, Figure 11, Figure 12, Figure 13, Figure 14 and Figure 15. Again, crisis phenomena appear where the system suddenly enters to, or exists from, chaos. In addition, for some values of the parameter *r*, the phenomenon of antimonotonicity appears, where the system enters chaos through a period doubling route and also exits from chaos following a reverse period halving route (Figure 13 and Figure 14). Lastly, the phenomenon of constant chaos appears when the bifurcation parameter is set to r=4 (Figure 15). This means that the system will be chaotic for all values of the triangular fuzzy number chosen.

In addition, Figure 16 and Figure 17 give a closer look at the bifurcation diagrams with respect to *z*, for r=3.98 and r=1.4, where the chaotic regions are interrupted by small windows of periodic behavior.

## 4. Application to Random Bit Generation

To showcase the high implementability and robustness of the proposed chaotic system, the application to random bit generation is considered. The proposed chaotic random bit generator is created by taking the value of $x_{i}$ in each iteration, discarding its first 10 decimal digits, and then comparing the resulting number with a threshold value, chosen here as 0.5. The bit value 1 is produced if the number is greater or equal to the threshold, and the bit value 0 is produced if it is not. Thus, the RBG tactic has the following form:(5)$$d_{i}=10^{10}x_{i}\;\mathrm{mod}\;1$$
(6)$$b_{i}=\left\{\begin{array}{cc} 1, & \mathrm{if}\;d_{i}\geq 0.5\\ 0, & \mathrm{if}\;d_{i}<0.5. \end{array}\right.$$

To test if the generated sequence is truly random, the FIPS (Federal Information Processing Standards) tests of the National Institute of Standards and Technology (NIST 800-22) are used [49]. All 15 given tests are considered. Each test results in a *p*-value, for which if p≥a the test is considered successful, where *a* is the level of significance chosen.

The tests were applied to 20 sequences of 1,000,000 bits each, for a=0.01, generated for three different parameter values: (r=4,z=0.5), (r=2,z=0.3), and (r=3.4,z=0.8). The results are shown in Table 1, Table 2 and Table 3, where it is seen that the sequence passes all tests for all three cases. For tests that have multiple case runs, the result of the last run is printed. The choice of multiple parameter values further showcases the versatility of the map.

In addition, Figure 18 shows the sensitivity of the system to initial conditions and parameters. It is seen that small changes lead to different trajectories and bit sequence after a very short number of iterations. In addition, Figure 19 depicts the autocorrelation and cross-correlation plots for a bit sequence of length $10^{4}$, generated for parameter values (r=4,z=0.5). For random sequences, the auto-correlation should have a delta like form, and the cross-correlation should be close to zero [5,13]. This is indeed verified. For the cross-correlation, two random bit sequences were generated, where the initial conditions of the two chaotic maps used are chosen as $x_{0}$ and $y_{0}=x_{0}+10^{-16}$. Finally, Figure 20 depicts the percentage of 1s in the bit sequence, with respect to the sequence length. This diagram shows that there exits 0–1 balancedness in the sequence, which is another desired property.

As for the key space, it is known that a system should have a key space larger than $2^{100}$ to resist brute force attacks [51], although some more recent works require a lower bound of $2^{128}$ [5,38,48]. The proposed system has two key parameters *r* and *z*, as well as the initial value $x_{0}$. Assuming z=4, for which the system exhibits constant chaos, the key space for a 16-digit accuracy is $10^{16}$ × $10^{16}=10^{32}\approx {\left(10^{3}\right)}^{10.6}\approx {\left(2^{10}\right)}^{10.6}=2^{106}$, which is higher than the minimum value $2^{100}$, but lower than $2^{128}$. An upper bound for the key space though can be computed by considerinrg the full spectrum of both parameters, which gives $10^{16}$×$10^{16}$×$10^{16}\approx 2^{160}$. Since the system is not chaotic for every combination of r,z, the real key space is between these two values.

## 5. Application to Image Encryption

The proposed RBG is further applied to the problem of image encryption, using the method proposed in [4]. The design consists of the following steps:Step 1.An m×n grayscale image is read as a matrix whose elements represent the gray value of each pixel, taking integer values in 0-255. The values are then converted to binary numbers and the matrix columns are reshaped to a single row vector *A*.Step 2.The resulting binary row vector *A* is combined with a binary vector *B* of equal length produced by the proposed RBG using the XOR command, resulting in the encrypted message C=A⊕B.Step 3.The encrypted sequence *C* can be transmitted safely and the original image can be reconstructed at the receiver end by taking A=C⊕B and following the reverse procedure of Step 1.

The above procedure is showcased using a 512×512 image of Lenna. The original image and the encrypted and decrypted ones are shown in Figure 21.

In order to study the security of the encryption design, several tests are performed on the encrypted image. First, the histograms of the original and encrypted images are computed, as shown in Figure 22. In contrast to the original image, the encrypted image has a uniform histogram, which makes it strong against statistical attacks.

Next, the correlation between adjacent vertical, horizontal and diagonal pixels is tested for the original and encrypted images and the results are shown in Figure 23. Here, 10000 randomly selected pairs of adjacent pixels are taken, and it is seen that in contrast to the original image, adjacent pixels in the encrypted one are uncorrelated. This is also verified by the values of the correlation coefficient γ shown in Table 4, were it is observed that for the encrypted image the value is close to zero. The correlation coefficient is computed using the following formulas:(7)$$\begin{array}{c} E\left(x\right)=\frac{1}{N}\sum_{i=1}^{N}x_{i}, \end{array}$$(8)$$\begin{array}{c} D\left(x\right)=\frac{1}{N}\sum_{i=1}^{N}{\left(x_{i}-E\left(x\right)\right)}^{2}, \end{array}$$(9)$$\begin{array}{c} cov\left(x,y\right)=\frac{1}{N}\sum_{i=1}^{N}\left(x_{i}-E\left(x\right)\right)\left(y_{i}-E\left(y\right)\right), \end{array}$$(10)$$\begin{array}{c} \gamma \left(x,y\right)=\frac{cov(x,y)}{\sqrt{D(x)}\sqrt{D(y)}}, \end{array}$$
where x,y are the gray values of two adjacent pixels, and *N* the number of adjacent pairs of pixels considered.

Finally, the information entropy is computed, which measures the randomness of a given signal, given by
(11)$$H\left(S\right)=-\sum_{i=0}^{N-1}p\left(s_{i}\right)\log_{2}p\left(s_{i}\right),$$
where $p(s_{i})$ is the possibility of appearance for the symbol $s_{i}$. The information entropy of an encrypted image should be close to 8. The information entropy of the original image is 7.4450, while for the encrypted image is 7.9670, so the encrypted image has a value closer to 8, which means that the information signal is safer against entropy attacks.

## 6. Conclusions

In this work, the logistic map was modified through the use of fuzzy triangular numbers, to give a new modified logistic map that exhibits a plethora of chaos related phenomena, for different parameter values. It was shown that the modified logistic map also reaches a higher Lyapunov exponent compared to the logistic map. The map was then applied to the problem of random bit generation, yielding positive results. It is important to note that this simple technique to increase the complexity of a chaotic map can easily be implemented to many other chaotic systems. In addition, using different fuzzy numbers other than triangular can lead to many modifications of this technique. These are both promising research topics for future works.

## Figures and Tables

**Figure 1 entropy-22-00474-f001:**
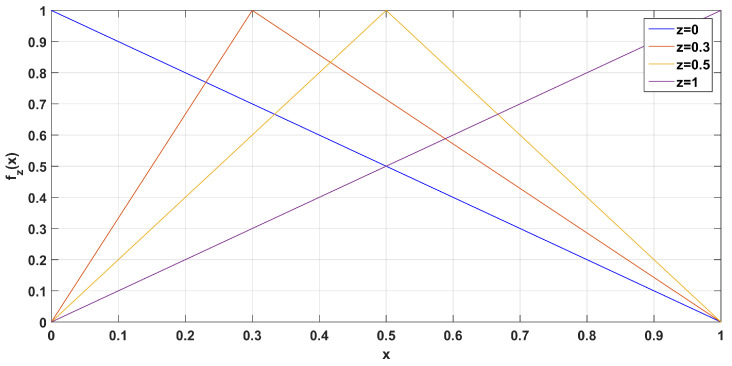
Examples of fuzzy trigonometric numbers for z=0,0.3,0.5,1.

**Figure 2 entropy-22-00474-f002:**
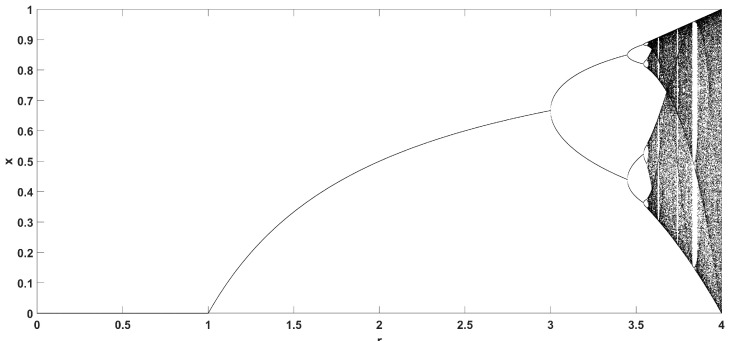
Bifurcation diagram of the logistic map.

**Figure 3 entropy-22-00474-f003:**
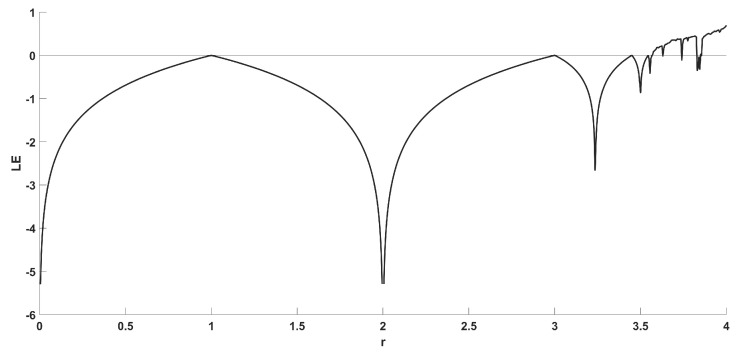
Diagram of Lyapunov exponent of the Logistics map.

**Figure 4 entropy-22-00474-f004:**
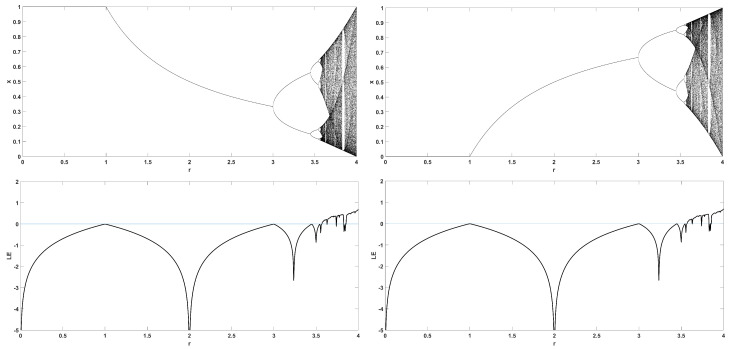
Bifurcation diagram of *x* versus the bifurcation parameter *r* and corresponding diagram of Lyapunov exponent for z=0 (**left**) and z=1 (**right**).

**Figure 5 entropy-22-00474-f005:**
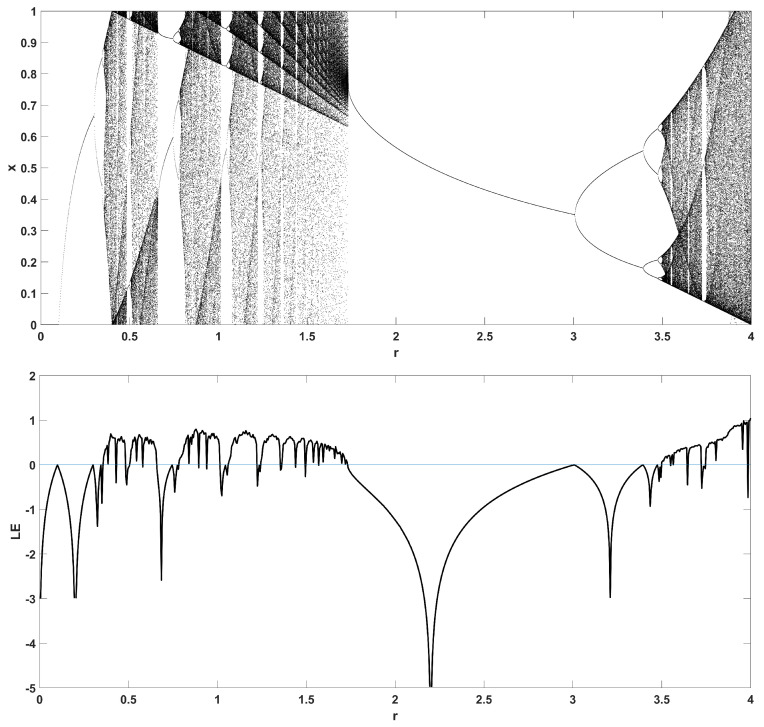
Bifurcation diagram of *x* versus the bifurcation parameter *r* and corresponding diagram of Lyapunov exponent for z=0.1.

**Figure 6 entropy-22-00474-f006:**
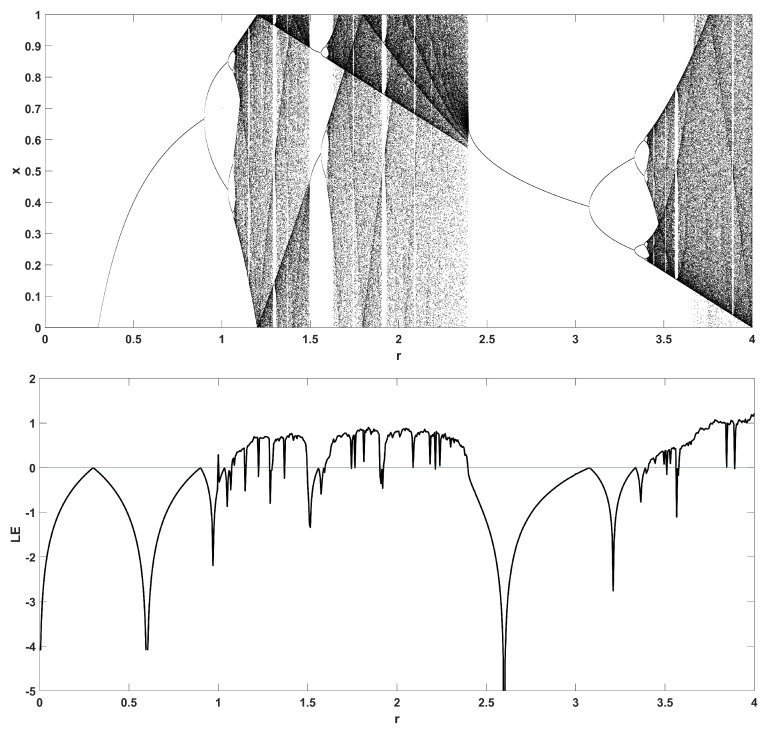
Bifurcation diagram of *x* versus the bifurcation parameter *r* and corresponding diagram of Lyapunov exponent for z=0.3.

**Figure 7 entropy-22-00474-f007:**
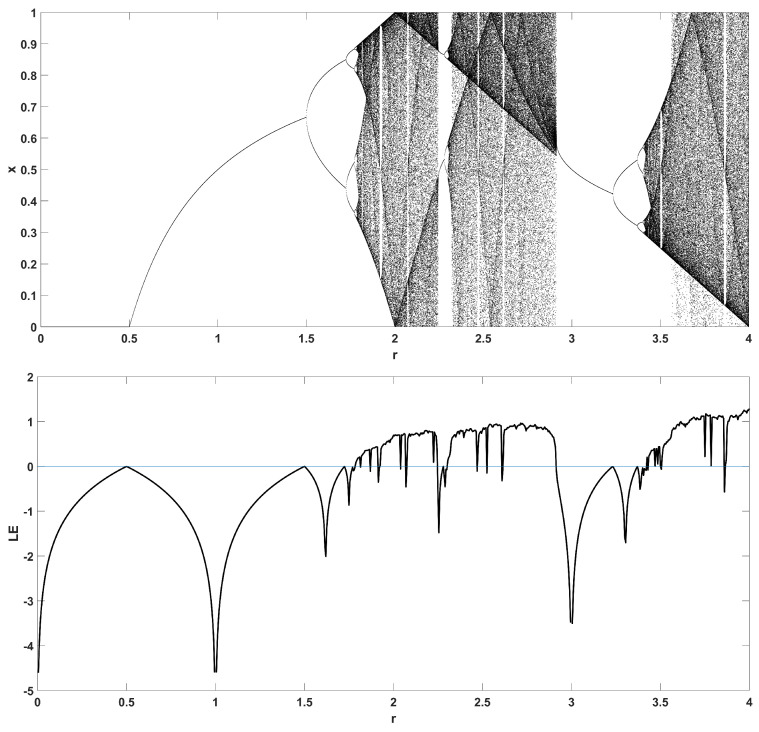
Bifurcation diagram of *x* versus the bifurcation parameter *r* and corresponding diagram of Lyapunov exponent for z=0.5.

**Figure 8 entropy-22-00474-f008:**
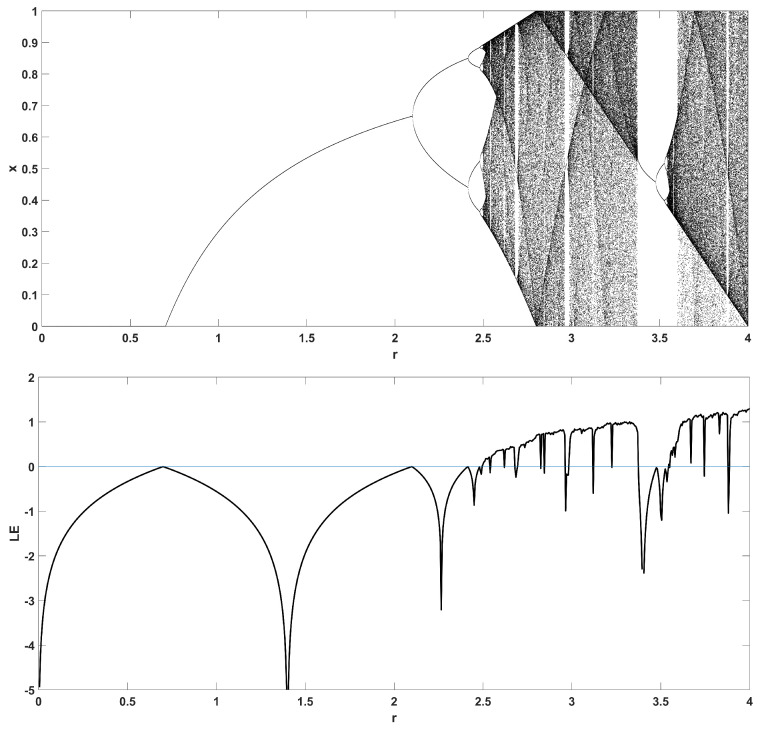
Bifurcation diagram of *x* versus the bifurcation parameter *r* and corresponding diagram of Lyapunov exponent for z=0.7.

**Figure 9 entropy-22-00474-f009:**
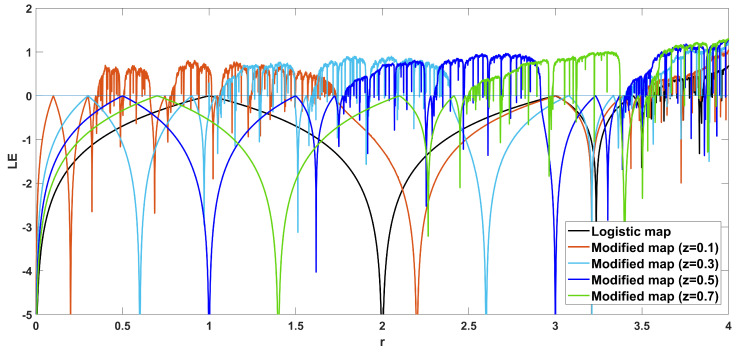
Diagram of Lyapunov exponents.

**Figure 10 entropy-22-00474-f010:**
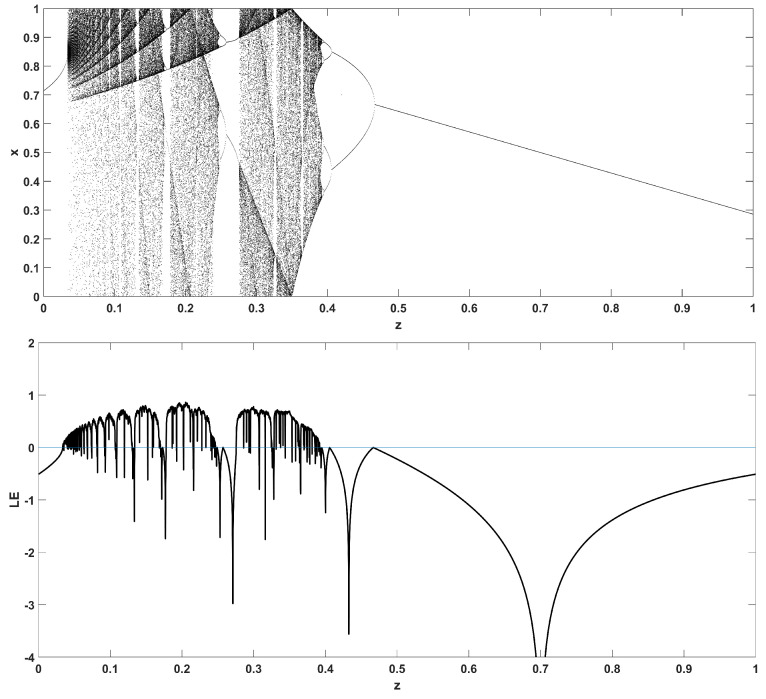
Bifurcation diagram of *x* versus the bifurcation parameter *z* and corresponding diagram of Lyapunov exponent for r=1.4.

**Figure 11 entropy-22-00474-f011:**
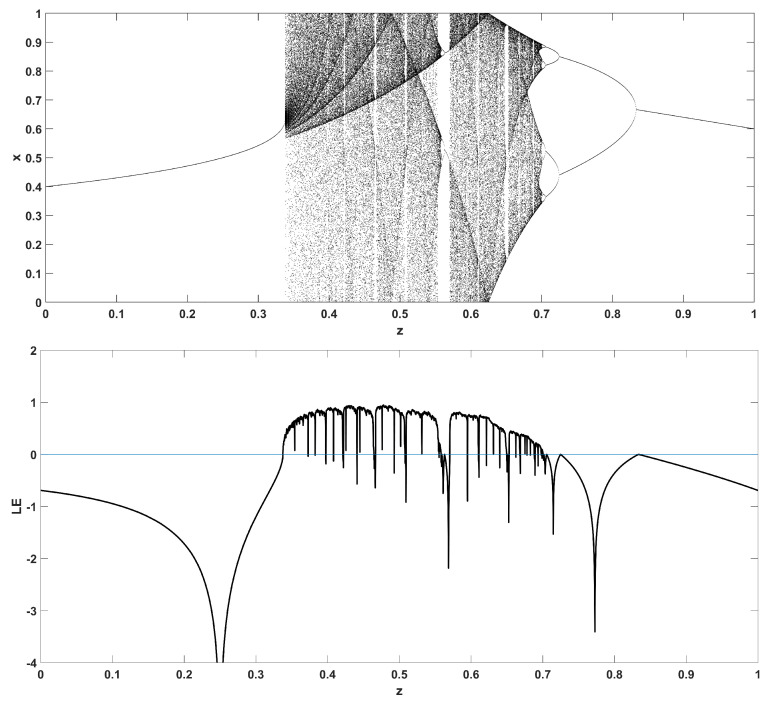
Bifurcation diagram of *x* versus the bifurcation parameter *z* and corresponding diagram of Lyapunov exponent for r=2.5.

**Figure 12 entropy-22-00474-f012:**
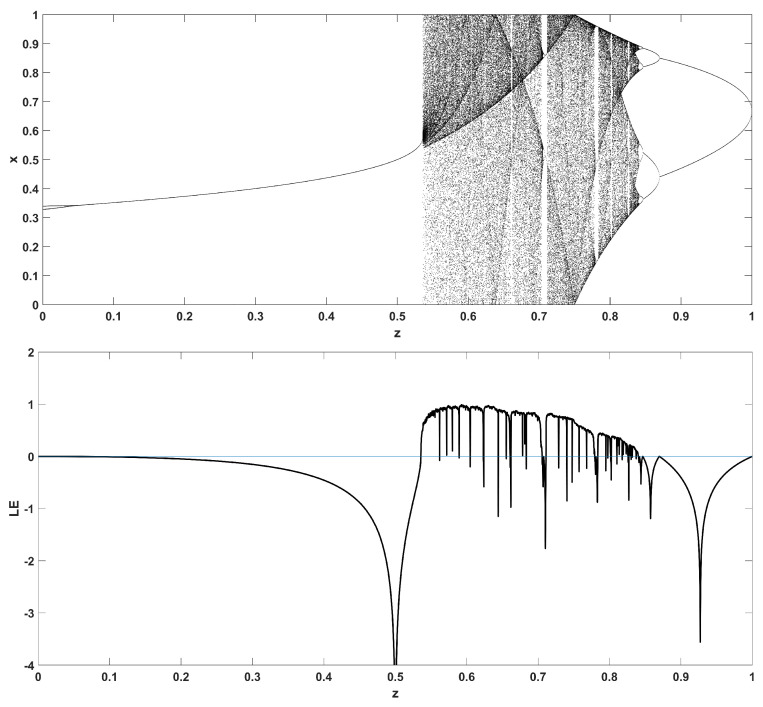
Bifurcation diagram of *x* versus the bifurcation parameter *z* and corresponding diagram of Lyapunov exponent for r=3.

**Figure 13 entropy-22-00474-f013:**
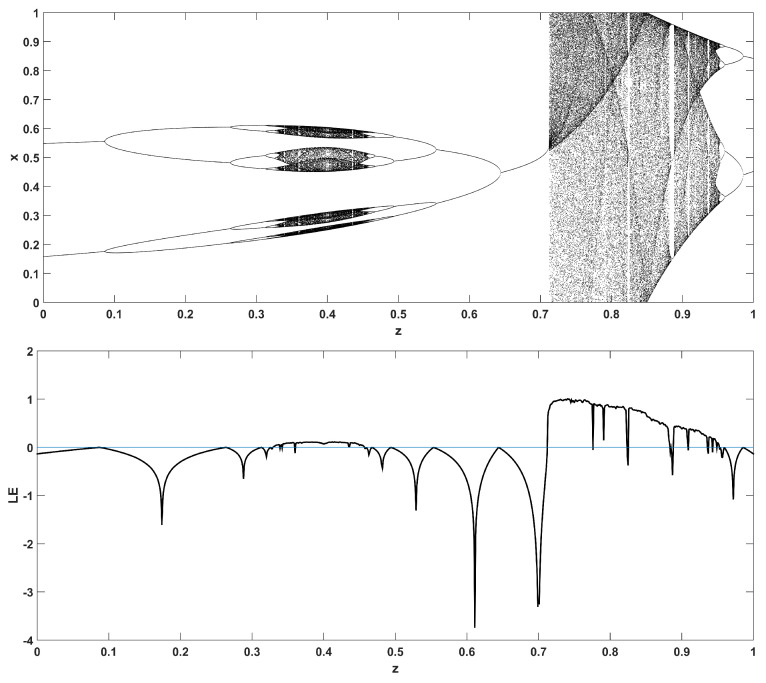
Bifurcation diagram of *x* versus the bifurcation parameter *z* and corresponding diagram of Lyapunov exponent for r=3.4.

**Figure 14 entropy-22-00474-f014:**
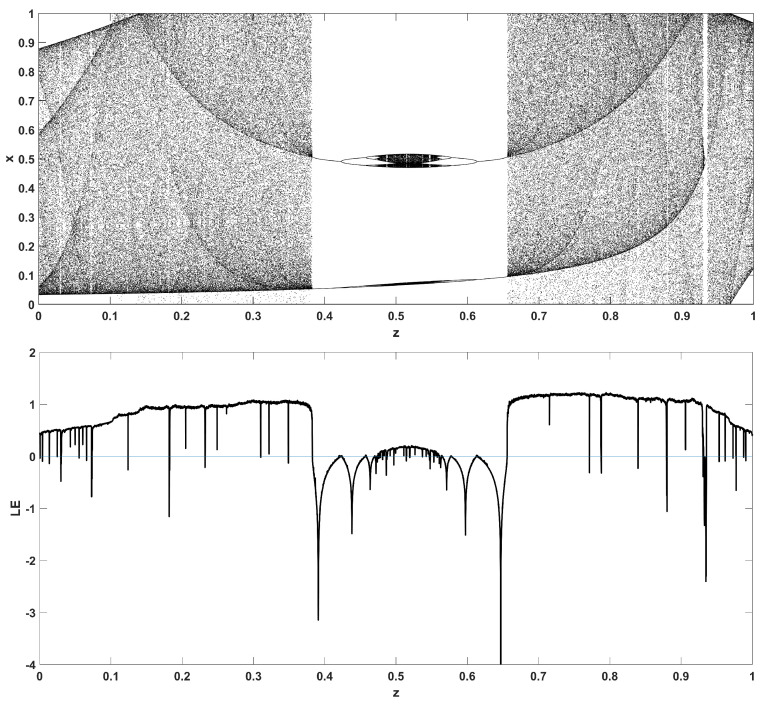
Bifurcation diagram of *x* versus the bifurcation parameter *z* and corresponding diagram of Lyapunov exponent for r=3.87.

**Figure 15 entropy-22-00474-f015:**
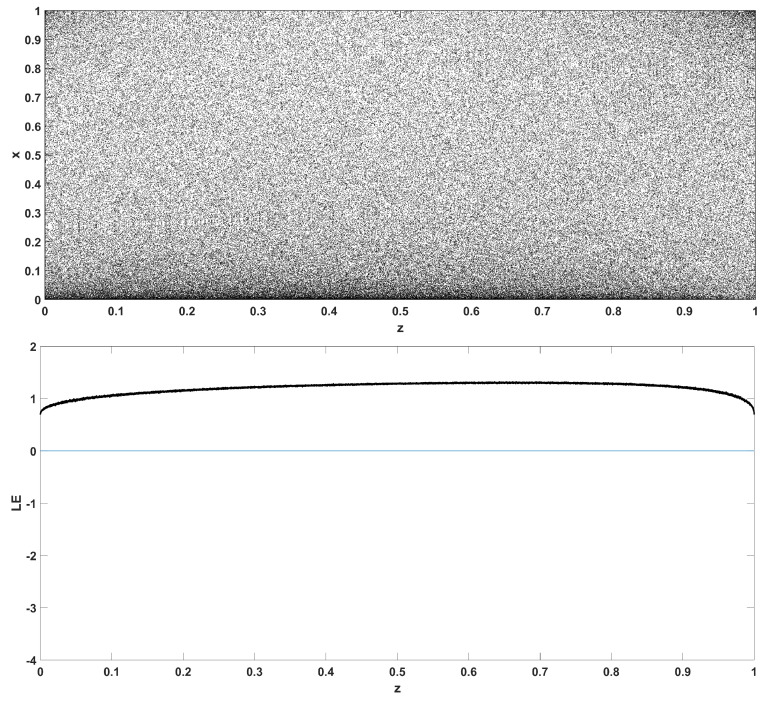
Bifurcation diagram of *x* versus the bifurcation parameter *z* and corresponding diagram of Lyapunov exponent for r=4.

**Figure 16 entropy-22-00474-f016:**
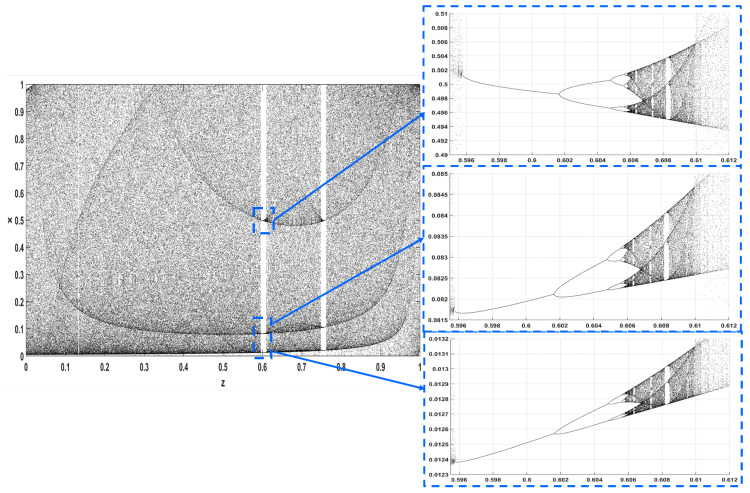
Bifurcation diagram of (Equation 4) with respect to parameter *z* for r=3.98.

**Figure 17 entropy-22-00474-f017:**
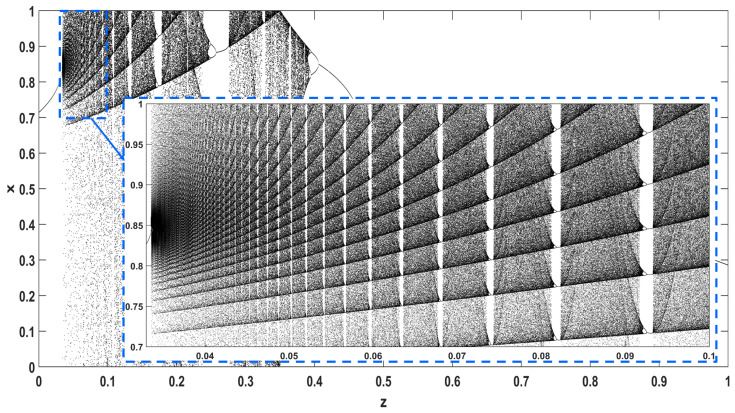
Bifurcation diagram of (Equation 4) with respect to parameter *z* for r=1.4.

**Figure 18 entropy-22-00474-f018:**
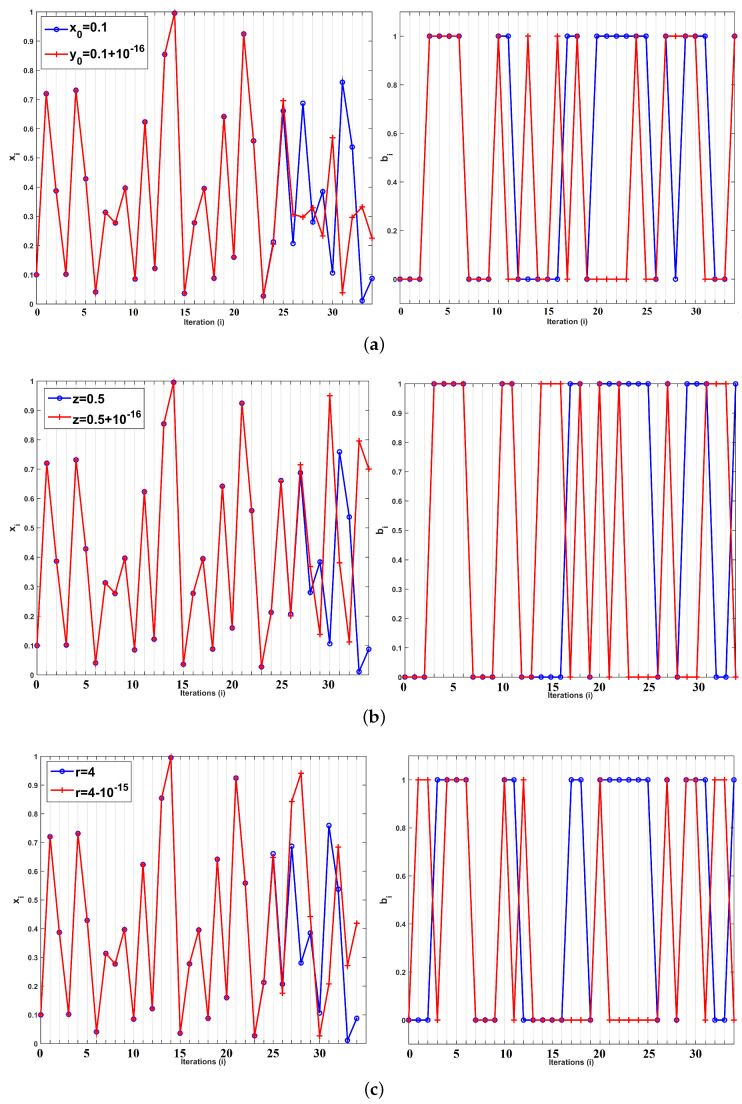
Sensitivity to initial conditions and parameter changes for (**a**) different initial conditions (r=4,z=0.5), (**b**) different *z*, $\left(x_{0}=y_{0}=0.1,r=4\right)$, and (**c**) different *r*, $\left(x_{0}=y_{0}=0.1,z=0.5\right)$.

**Figure 19 entropy-22-00474-f019:**
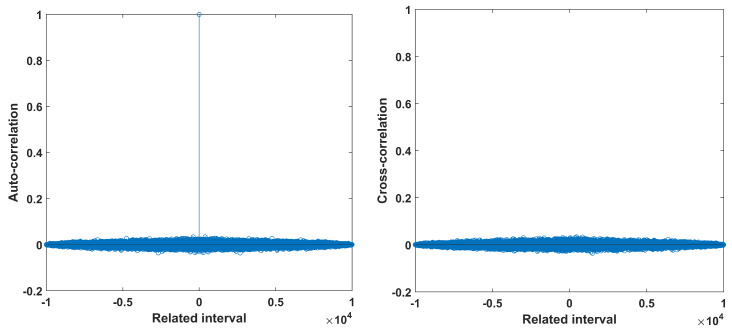
Auto-correlation and cross-correlation of the proposed RBG, for (r=4,z=0.5).

**Figure 20 entropy-22-00474-f020:**
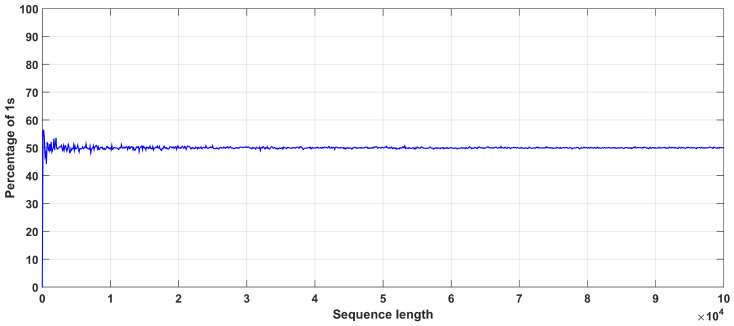
Occurrence of 1’s in the sequence the proposed PRBG, for (r=4,z=0.5).

**Figure 21 entropy-22-00474-f021:**
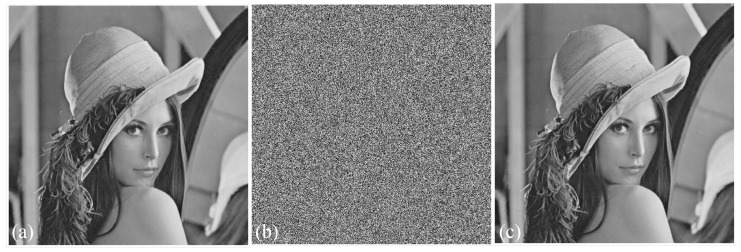
(**a**) Original image, (**b**) encrypted, and (**c**) decrypted.

**Figure 22 entropy-22-00474-f022:**
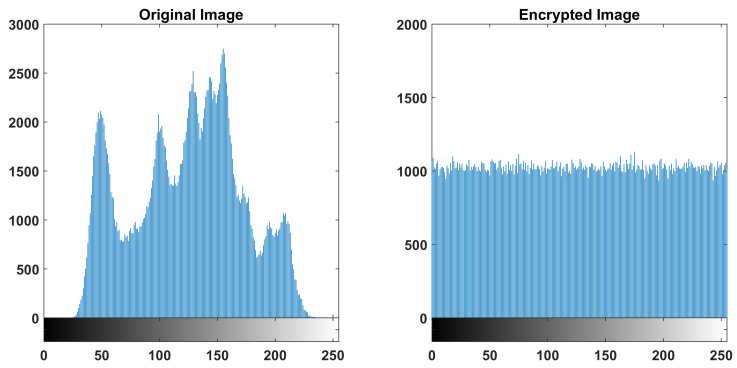
Histograms of the plain and encrypted image.

**Figure 23 entropy-22-00474-f023:**
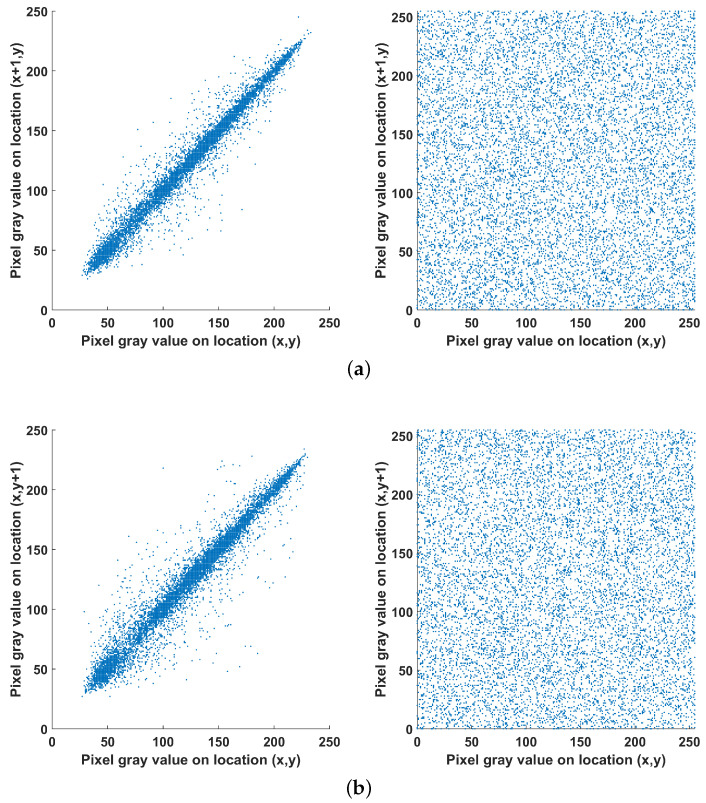
Correlation analysis of two (**a**) horizontal, (**b**) vertical and (**c**) diagonal adjacent pixels for the original (**left**) and encrypted (**right**) image.

**Table 1 entropy-22-00474-t001:** National Institute of Standards and Technology (NIST) statistical test results, with a=0.01, and r=4,z=0.5.

If p≥α, the Test Is Successful				
**No.**	**Statistical Test**	***p*** **-Value**	**Proportion**	**Result**
1	Frequency	0.437274	20/20	success
2	Block Frequency	0.964295	20/20	success
3	Cumulative Sums	0.534146	20/20	success
4	Runs	0.911413	20/20	success
5	Longest Run	0.534146	19/20	success
6	Rank	0.834308	19/20	success
7	FFT	0.534146	20/20	success
8	Non-Overlapping Template	0.534146	20/20	success
9	Overlapping Template	0.534146	19/20	success
10	Universal	0.964295	19/20	success
11	Approximate Entropy	0.534146	19/20	success
12	Random Excursions	0.911413	10/10	success
13	Random Excursions Variant	0.066882	10/10	success
14	Serial	0.437274	20/20	success
15	Linear Complexity	0.964295	20/20	success

**Table 2 entropy-22-00474-t002:** NIST statistical test results, with a=0.01, and r=2,z=0.3.

If p≥α, the Test Is Successful				
**No.**	**Statistical Test**	***p*** **-Value**	**Proportion**	**Result**
1	Frequency	0.637119	19/20	success
2	Block Frequency	0.122325	20/20	success
3	Cumulative Sums	0.090936	19/20	success
4	Runs	0.964295	20/20	success
5	Longest Run	0.350485	20/20	success
6	Rank	0.350485	20/20	success
7	FFT	0.275709	20/20	success
8	Non-Overlapping Template	0.066882	20/20	success
9	Overlapping Template	0.739918	20/20	success
10	Universal	0.834308	20/20	success
11	Approximate Entropy	0.275709	20/20	success
12	Random Excursions	0.437274	11/11	success
13	Random Excursions Variant	0.637119	11/11	success
14	Serial	0.637119	19/20	success
15	Linear Complexity	0.090936	20/20	success

**Table 3 entropy-22-00474-t003:** NIST statistical test results, with a=0.01, and r=3.4,z=0.8.

If p≥α, the Test Is Successful				
**No.**	**Statistical Test**	***p*** **-Value**	**Proportion**	**Result**
1	Frequency	0.275709	18/20	success
2	Block Frequency	0.437274	20/20	success
3	Cumulative Sums	0.637119	19/20	success
4	Runs	0.275709	20/20	success
5	Longest Run	0.122325	20/20	success
6	Rank	0.437274	20/20	success
7	FFT	0.090936	20/20	success
8	Non-Overlapping Template	0.213309	20/20	success
9	Overlapping Template	0.834308	19/20	success
10	Universal	0.437274	20/20	success
11	Approximate Entropy	0.004301	20/20	success
12	Random Excursions	0.035174	14/14	success
13	Random Excursions Variant	0.066882	14/14	success
14	Serial	0.437274	20/20	success
15	Linear Complexity	0.964295	20/20	success

**Table 4 entropy-22-00474-t004:** Correlation coefficients of two adjacent pixels in the original and encrypted image.

	Original	Encrypted
Horizontal	0.9843	0.0046
Vertical	0.9724	0.0063
Diagonal	0.9573	0.0023

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
