# Peer review of "Modification of the Logistic Map Using Fuzzy Numbers with Application to Pseudorandom Number Generation and Image Encryption"

_entropy, 2020, doi:10.3390/e22040474_

Round 1

Reviewer 1 Report

The main question addressed by research is an image encryption application, it is interesting.
The subject is original at 75%, it is an improvement of the logistics map using fuzzy numbers, it is well written, clear and readable.
The conclusions are almost consistent with the evidence and arguments presented, there are questions that remain unanswered and that we can find answers in the future such as the complexity of the calculations and the effectiveness of this type of encryption compared to other existing in literature.

Accept after minor revision (corrections to minor methodological errors and text editing)

Author Response

Thank you very much for your comments.

Indeed, a future extension of this work would be the application of the proposed technique to more complex methods, as well as the use in different 1d maps.

We have also proofread the manuscript again and fi xed typos and expressions.

Reviewer 2 Report

Problem aimed in this paper is simple: authors introduce fuzzy numbers into difference equation of well known logistic map.

I appreciate high-resolution bifurcation diagrams for different values of fuzzy trigonometric numbers. Achieved results show that properly modified logistic map can exhibit larger Lyapunov exponent than original formula, i.e. map becomes more chaotic. Figure 9 nicely shows that chaotic windows change position with respect to r and z. Also, NIST statistical tests performed for promising combinations of parameter r and z was successfull.

I really enjoyed reading this manuscript. However, I strongly recommend to enlarge axis descriptions of Fig. 16 (zoomed diagrams) and Fig. 18.

Check your paper against minor typing mistakes. For example, improtant -> important in Conclusions.

Author Response

Thank you for your comments.

Indeed we have proofread the manuscript again.

Also, regarding the fi gures, in Fig. 16 we changed the size of the subfi gures to make them larger.

Also in Fig. 18 we changed the x-axis labels to make them clearer.

Reviewer 3 Report

Proposed modification of the classic logistic map using fuzzy triangular numbers is an interesting way to obtain higher complexity than original classic logistic map. Authors present results in application to the problem of pseudo random bit generation, using a simple rule to generate the bit sequence, and then to to the problem of the image encryption. All the results are clearly presented in a wide range of the system parameters, showing stability of the obtained results. Final presentation of the image decription and encryption is very very convincing. With no doubt presented work is on high scientific level and worth publication.

No problems, that should be corrected or better explained were found

Author Response

Thank you for your positive comments.